# Large-scale characterization of cocaine addiction-like behaviors reveals that escalation of intake, aversion-resistant responding, and breaking-points are highly correlated measures of the same construct

Giordano de Guglielmo[1]\*[†], Lieselot Carrette[1†], Marsida Kallupi[1], Molly Brennan[1], Brent Boomhower[1], Lisa Maturin[1], Dana Conlisk[2], Sharona Sedighim[1], Lani Tieu[1], McKenzie J Fannon[1], Angelica R Martinez[1], Nathan Velarde[1], Dyar Othman[1], Benjamin Sichel[1], Jarryd Ramborger[1], Justin Lau[1], Jenni Kononoff[2], Adam Kimbrough[1], Sierra Simpson[1], Lauren C Smith[1,2], Kokila Shankar[1,2], Selene Bonnet-Zahedi[1,3], Elizabeth A Sneddon[1], Alicia Avelar[1,2], Sonja Lorean Plasil[1], Joseph Mosquera[1], Caitlin Crook[1], Lucas Chun[1], Ashley Vang[1], Kristel K Milan[1], Paul Schweitzer[1], Bonnie Lin[1], Beverly Peng[1], Apurva S Chitre[1], Oksana Polesskaya[1,4], Leah C Solberg Woods[5], Abraham A Palmer[1,4], Olivier George[1]\*

[1]Department of Psychiatry, University of California, San Diego, La Jolla, United States; [2]Department of Neuroscience, The Scripps Research Institute, La Jolla, San Diego, United States; [3]Institut de Neurosciences de la Timone, Aix-Marseille Université, Marseille, France; [4]Institute for Genomic Medicine, University of California, San Diego, La Jolla, United States; [5]Department of Internal Medicine, Section on Molecular Medicine, Wake Forest University School of Medicine, Winston-Salem, United States

\*For correspondence: gdeguglielmo@health.ucsd.edu (GdG); olgeorge@health.ucsd.edu (OG)

[†]These authors contributed equally to this work

Competing interest: The authors declare that no competing interests exist.

## eLife Assessment

This manuscript tackles a significant problem in addiction science: how interdependent are measures of "addiction-like" behavioral phenotypes? The manuscript provides **compelling** evidence that, under these experimental conditions, escalation of intake, punishment-resistant responding, and progressive ratio break points reflect a single underlying construct rather than reflect distinct unrelated measures. The exceptionally large sample size and incorporation of multiple behavioral endpoints add strength to this paper, and make it an **important** resource for the field.

**Abstract** Addiction is commonly characterized by escalation of drug intake, compulsive drug seeking, and continued use despite harmful consequences. However, the factors contributing to the transition from moderate drug use to these problematic patterns remain unclear, particularly regarding the role of sex. Many preclinical studies have been limited by small sample sizes, low genetic diversity, and restricted drug access, making it challenging to model significant levels of intoxication or dependence and translate findings to humans. To address these limitations, we

characterized addiction-like behaviors in a large sample of >500 outbred heterogeneous stock (HS) rats using an extended cocaine self-administration paradigm (6 hr/daily). We analyzed individual differences in escalation of intake, progressive ratio (PR) responding, continued use despite adverse consequences (contingent foot shocks), and irritability-like behavior during withdrawal. Principal component analysis showed that escalation of intake, progressive ratio responding, and continued use despite adverse consequences loaded onto a single factor that was distinct from irritability-like behaviors. Categorizing rats into resilient, mild, moderate, and severe addiction-like phenotypes showed that females exhibited higher addiction-like behaviors, with a lower proportion of resilient individuals compared to males. These findings suggest that, in genetically diverse rats with extended drug access, escalation of intake, continued use despite adverse consequences, and PR responding are highly correlated measures of a shared underlying construct. Furthermore, our results highlight sex differences in resilience to addiction-like behaviors.

## Introduction

Cocaine use disorder remains a serious public health problem in the USA, with 2.2 million regular cocaine users in 2019 and over a million individuals with cocaine use disorder in the past year (**SAMHSA, 2019**). The use of cocaine is associated with substantial morbidity and elevated rates of healthcare utilization (**Butler et al., 2017**), and over the last 10 years, the number of cocaine-related deaths has quadrupled (**Hedegaard et al., 2018**). A key unanswered question for addiction research remains why casual patterns of drug consumption escalate to problematic patterns associated with high motivation and a compulsive-like pattern of drug use in some individuals, but not others and how sex may affect this trajectory (**George and Koob, 2017**; **Piazza and Deroche-Gamonet, 2013**). Although addiction affects both males and females, most preclinical studies have focused on males (**Beery and Zucker, 2011**; **Shansky and Murphy, 2021**). Even in studies that use males and females, low sample size often makes it difficult to draw reliable conclusions about the role of sex differences in addiction vulnerability (**Becker and Koob, 2016**; **Carroll and Lynch, 2016**). To investigate the critical questions about how sex and individual differences interact to produce the vulnerability to develop addiction-like large sample sizes are urgently needed.

Recently, the preclinical addiction field has moved from recognizing 'compulsive drug seeking/ use' and 'continued seeking/use despite negative consequences' as two distinct aspects of addiction, to examining compulsive-like behavior nearly exclusively by models of continued seeking/use despite negative consequences (**Chen et al., 2013**; **Domi et al., 2021**; **Siciliano et al., 2019**; **Timme et al., 2022**). The three most prevalent behavioral measures used in animal models of addiction are an escalation of drug intake, increased motivation under a PR schedule of reinforcement, and continued drug use despite adverse consequences, such as a footshock. These measures are thought to capture different aspects of addiction-like behaviors. Some researchers argue that continued drug use despite adverse consequences is the most critical measure for identifying an addiction phenotype, as it reflects the compulsive nature of drug use (**Deroche-Gamonet et al., 2004**; **Vanderschuren and Everitt, 2004**). This view is supported by findings that responding despite adverse consequences is sometimes uncorrelated with drug taking/seeking (**Belin et al., 2008**; **Chen et al., 2013**; **Domi et al., 2021**; **Pelloux et al., 2007**; **Siciliano et al., 2019**; **Timme et al., 2022**), suggesting that it may measure a distinct psychological construct. Unfortunately, most of these studies have used low sample sizes and used animal models with limited access to the drug that are often not associated with significant levels of intoxication or dependence, making it difficult to draw definitive conclusions.

To address these issues, we analyzed addiction-like behaviors in >500 male and female rats that were phenotyped for an ongoing gene-wide association study. We used HS rats because they are the most highly recombinant rat intercross available and exhibit remarkable individual differences in addiction-like behaviors (**Carrette et al., 2021**; **Carrette et al., 2022**; **Chitre et al., 2020**; **Duttke et al., 2022**; **Kallupi et al., 2020**; **Sedighim et al., 2021**; **Solberg Woods and Palmer, 2019**). HS rats were created by interbreeding eight inbred strains and maintaining them as an outbred population in a way that minimizes inbreeding, thereby maximizing genetic diversity within the colony (**Solberg Woods and Palmer, 2019**). We used extended access to intravenous cocaine self-administration and characterized escalation of cocaine intake, breaking point under a progressive ratio schedule of

reinforcement, continued use despite adverse consequences (footshock), and irritability-like behavior as a measure of negative affective state during withdrawal.

## Results

### Evaluation of individual differences in addiction-like behaviors

We assessed addiction-like behaviors in HS rats self-administering cocaine according to the standardized protocol shown in *Figure 1A*. Animals were tested in 12 cohorts of 46–60 rats per cohort. Thirty-four animals were excluded from the analysis because they lost their catheter patency (failed Brevital test) before the end of the behavioral protocol. After catheterization surgery and recovery, the rats were trained to self-administer cocaine in 10 daily 2 hr short access (ShA) sessions and then in 14 daily 6 hr long access (LgA) sessions. Over the course of the extended access to cocaine self-administration phase, the animals showed an escalation of intake, as measured by the significant increase of cocaine rewards taken per hour from day 3 of LgA onwards vs the first day of LgA (Fig. 1B, $F_{(566,7358)} = 124.4$; p<0.0001 after one-way ANOVA, followed by Bonferroni post hoc comparisons). Evolution of total intake is given in the supporting material *Figure 1—figure supplement 1A*. The number of cocaine infusions in the last three days of LgA were averaged for each rat in order to compare the levels of cocaine intake at the end of the behavioral protocol in the whole population (*Figure 1C*). The results showed a bimodal distribution with ~20% of rats maintaining very low levels of intake, defined as <50 infusions per session (that corresponded to ~8 infusions/hr, levels normally achieved during ShA, see *Figure 1B*) and 80% of the rats escalating their cocaine intake over the course of the extended access protocol. Motivation for cocaine was tested both after ShA and LgA, using a PR test (*Figure 1D*). The number of cocaine infusions significantly increased after extended access to cocaine self-administration, compared to the levels at the end of the short access phase ($F_{2,1690} = 135.5$; p<0.0001) with the population following a bimodal distribution. Compulsive-like behavior was tested by pairing the cocaine reward with a foot shock (0.3 mA, 30% contingency). HS rats showed high individual variability in compulsive-like cocaine intake with a general decrease in responding during the shock session compared to the preshock session ($t_{465}=17.88$, p<0.0001). Correlational analysis showed that the lever pressing during the preshock session strongly predicted responding during the shock session. This result was true for the whole population ($R=0.58$, p<$2.2 \times 10^{-16}$) as well as for each individual cohort (*Figure 1F*). Cocaine-induced withdrawal was assessed by measuring irritability-like behavior using the bottle brush test (*Kimbrough et al., 2017*). Irritability-like behavior was measured at the beginning and at the end of the behavioral paradigm (pre- and post-cocaine, *Figure 1G*) and the differences from baseline were compared to data obtained from age-matched naïve animals. Irritability-like behavior was increased in cocaine rats during withdrawal compared to naïve animals as demonstrated by the increased total irritability score ($t_{427}=7.57$; p<0.0001 vs naïve after t-test). When breaking the total irritability score in defensive and aggressive responses, we found that animals with a history of cocaine increased their aggressive ($t_{427}=7.82$; p<0.0001 vs naïve after t-test) and defensive behaviors ($t_{427}=2.79$; p=0.0069 vs naïve after the t-test), compared to their naïve counterparts. Finally, to identify patterns in the multidimensional dataset of addiction-like behaviors, a principal component analysis (PCA) was used (*Figure 1H*). This method allows for the identification of latent variables - principal components - that can account for most of the variance within the data, thus simplifying complex data structures. Three significant findings emerged from this analysis. First, the majority of the variance was accounted for by the first two principal components (PC), with PC1 and PC2 explaining 48.6% and 15.7% of the variance, respectively. Second, measures related to short-term cocaine intake were orthogonal to those collected during longer access periods, showing a consistent daily shift from primarily loading onto PC1 to gradually loading onto PC2. Third, metrics associated with escalating intake, motivation, and compulsive-like responses (calculated by normalizing the behavioral data into indexes using *Z*-scores) were loaded in the same direction onto PC2.

### Sex differences in addiction-like behaviors

Major sex differences were observed with females acquiring cocaine self-administration faster and at higher levels than males under short access condition (2 hr/day, *Figure 2A*) and during the extended access (6 hr/day, *Figure 2A*). A two-way ANOVA with sex as between factor and sessions as within factor showed a significant effect of sex ($F_{(1, 565)}=54.87$; p<0.001), sessions ($F_{(23, 565)}=345.3$; p<0.001)

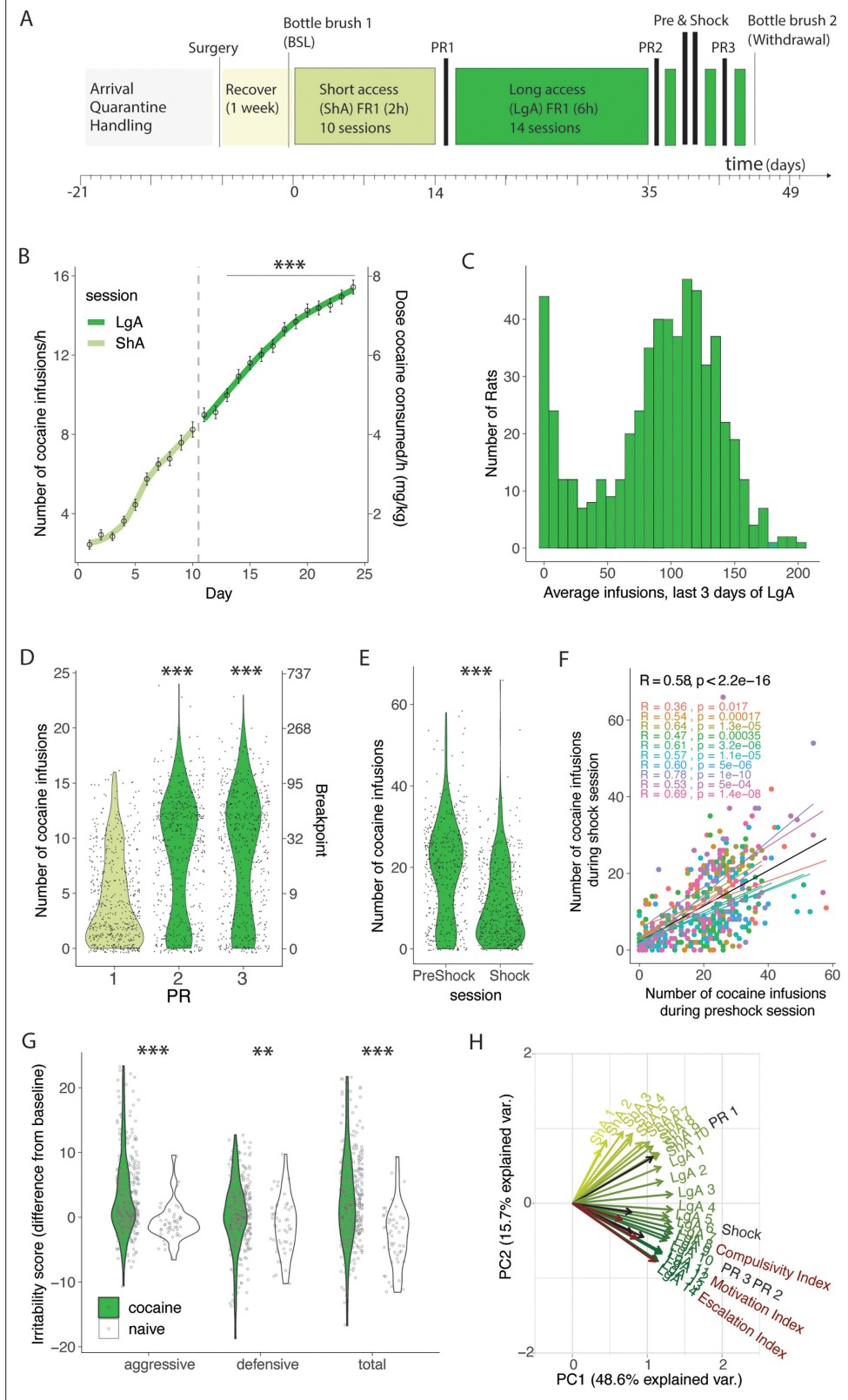

**Figure 1.** Individual differences in addiction behaviors in heterogeneous stock (HS) rats following intravenous cocaine self-administration. (**A**) Timeline of the behavioral paradigms. (**B**) Number of cocaine infusions in the first hour of cocaine self-administration during short (2 hr, short access: ShA) and long (6 hr, long access: LgA) access (N=567, *** p<0.0001 vs the first LgA session). (**C**) Average number of daily infusions for the last 3 days

*Figure 1 continued on next page*

*Figure 1 continued*

of LgA (N=567). (**D**) Violin plot of the number of cocaine infusions under progressive ratio (PR) tested after ShA (1) and LgA, before (2) and after (3) the Shock session (N=560, *** p<0.0001). (**E**) Number of infusions despite footshock after LgA compared to a 1 hr preshock session (N=466, *** p<0.0001). (**F**) Correlation of responding during the shock and preshock session is reproduced in each of the 10 color-coded cohorts (N=466, p<0.0001). (**G**) Difference in irritability scores after LgA and at baseline (N=380 + 49 naive, behavior ***p<0.0001, **p<0.001 vs naive) (**H**) Principal component analysis of cocaine infusions over all sessions ShA1-10, LgA1-14, PR1-3, Shock with escalation, motivation, and compulsivity indices.

The online version of this article includes the following source data and figure supplement(s) for figure 1:

**Source data 1.** Raw data utilized for generating *Figure 1*.

**Figure supplement 1.** Additional representations of the intake and lever presses are associated with *Figure 1B*.

**Figure supplement 1—source data 1.** Raw data utilized for generating *Figure 1—figure supplement 1*.

and of the interaction sex*sessions ($F_{(23, 12836)}$=5.542; p<0.001). The Bonferroni post hoc test demonstrated that females self-administered more cocaine compared to males starting from day 4 onward (p<0.0001). The Cohen d effect size was negligible at sha1 (0.09 [-0.10; 0.27]), small at lga1 (0.47 [0.31;0.64]), and small at lga14 (0.41 [0.25;0.58]). *Figure 2B* shows a bimodal distribution with some

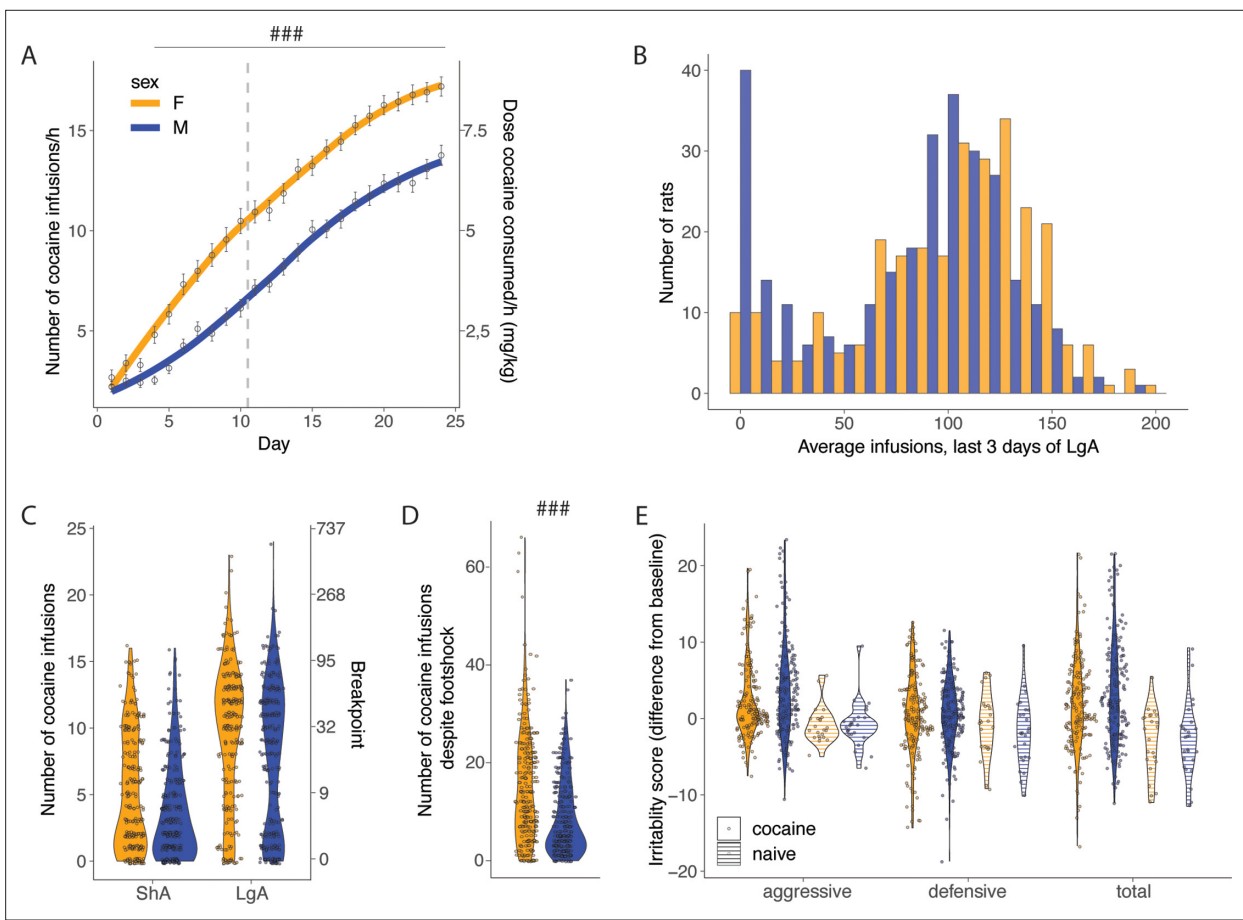

**Figure 2.** Sex differences in addiction-like behaviors. (**A**) Cocaine infusions during short (2 hr, short access: ShA) and long (6 hr, long access: LgA) access of cocaine self-administration (N=275 F and 292 M, ###p<0.0001 vs males) (**B**) Average number of daily infusions for the last 3 days of LgA in male and female rats. (**C**) Violin plot of a number of cocaine infusions under progressive ratio (PR) test after ShA and LgA (N=273 F and 291 M). (**D**) Number of infusions despite footshock after LgA (N=275 F and 292 M, ###p<0.0001). (**E**) Increase in irritability-like behavior in males and females after LgA compared to naive rats, no significant sex differences were detected (N=186 F+194 M+23 naive F+26 naive M).

The online version of this article includes the following source data for figure 2:

**Source data 1.** Raw data utilized for generating *Figure 2*.

rats maintaining very low levels of intake. The group of rats with very low intake was composed mostly of male rats. In the progressive ratio test, the two-way ANOVA with sex as between subjects factor and time as within subjects factor showed a significant effect of sex ($F_{(1, 558)}$=23.57; p<0.001), time ($F_{(1, 558)}$=332.65; p<0.001) but not a sex*time interaction ($F_{(1, 558)}$=0.35; p=NS), confirming that motivation for cocaine was increased in both sexes over time from ShA to LgA with a large effect size (Cohen d=0.86 [0.74;0.98]) and that females had significantly higher motivation for cocaine compared to males with a small effect size (Cohen d=0.32 [0.20;0.44]) independently from the length of access to cocaine (*Figure 2C*). Similar results with a medium effect size were found for compulsive-like behavior, with females obtaining more infusions despite adverse consequences ($t_{565}$=7.86; p<0.0001 after t-test, Cohen d=0.67 [0.50;0.84]; *Figure 2D*). We did not detect any sex-differences in irritability-like behavior (*Figure 2E*), as demonstrated by the non-significant sex factor in the two-way ANOVA ($F_{(1, 425)}$=0.78; p=NS, Cohen d=0.19 [0.00;0.38]). Males and females showed large increases in the total irritability scores, compared to naïve rats, as demonstrated by the significant effect of groups ($F_{(34, 64)}$=34.64; p<0.0001, Cohen d=0.89 [0.59;1.20]). Also when breaking the total irritability score in defensive and aggressive responses, we did not find significant sex effects (sex factor in two-way ANOVA, $F_{(1,854)}$ = 0.005; p=NS, Cohen d=0.09 [0.05;0.22]), but found large increases in the aggressive and defensive irritability scores between both sexes, compared to naive (group factor in two-way ANOVA, $F_{(1,854)}$ = 24.48; p<0.001, Cohen d=0.54 [0.32;0.75]). These results show that compared to the large effects of drug use, sex differences are present with small (for escalation and motivation, Cohen d~0.3–0.5) to medium (for compulsive-like behavior, Cohen d ~0.7) effect sizes.

## Individual differences in cocaine addiction-like behavior: Characterization of resilient and vulnerable rats

While all HS rats acquired cocaine self-administration during the ShA phase, we detected major individual differences in their escalation profile that resulted in a bimodal distribution of cocaine intake. We identified animals that were resilient (~20% of the entire population, they showed an hourly intake of less than 8 infusions/hr, similar to the intake during ShA, *Figure 1B and C*) or vulnerable (intake higher than 8 infusions/hr) to cocaine addiction-like behaviors (*Figure 3A and B*). From the analysis of sex differences above, we expected the Resilient group to contain more males. Indeed, among the resilient animals, there were twice as many males compared to females (N=122 total with 82 males and 40 females). The amounts in the vulnerable group were almost equal (N=445 total with 210 males and 235 females). Vulnerable rats took more cocaine as demonstrated by the significant interaction on the two-way ANOVA with groups as between factor and time as within factor ($F_{(23, 12835)}$=85.02; p<0.0001). The Bonferroni post hoc showed higher levels of cocaine intake in the vulnerable population starting from ShA day 3. When looking at the escalation patterns in vulnerable and resilient rats, we found that the vulnerable group showed an escalation of cocaine intake (significant groups*day interaction, $F_{(13, 7344)}$=50.16; p<0.0001) starting from day 3 of extended access (p<0.001 vs day 1, after two-way ANOVA followed by Bonferroni post hoc), while the rats in the resilient group did not show escalation of intake over the course of the behavioral paradigm, keeping the same level of cocaine infusions (*Figure 3A*). The Cohen d effect size was negligible at ShA1 (0.00 [-0.22; 0.22]), large at lga1 (0.84 [0.63;1.04]), and large at lga14 (2.97 [2.71;3.24]). Major differences between vulnerable and resilient rats were also observed in the motivation to obtain cocaine (PR, two-way ANOVA, significant group factor $F_{(1, 558)}$=249.17; p<0.0001, Cohen d=1.13 [0.98;1.28] and significant group* day interaction, $F_{(1, 558)}$=86.97; p<0.0001 followed by Bonferroni post hoc, *Figure 3C*) and in compulsive-like responding (foot shock, $t_{567}$=8.40 p<0.001, Cohen d=0.79 [0.58;0.99]; *Figure 3D*), but not in irritability-like behavior ($t_{378}$=1.17 p=NS, Cohen d=0.14 [-0.11;0.39]; *Figure 3E*). The difference in escalation of intake correlated with differences in the other addiction-like behaviors (R=0.6, p<0.001, for PR and R=0.37, p<0.001 for compulsivity), but did not correlate with irritability-like behavior (R=0.048, p=NS).

## Addiction Index: Evaluation of individual differences in addiction-like behaviors

To take advantage of all the behaviors related to compulsive intake and withdrawal and further identify subjects that are consistently vulnerable *vs.* resilient to compulsive cocaine use, each measure was normalized into an index using its Z-score ($Z = \frac{x-\mu}{\sigma}$), where $\chi$ is the raw value, μ is the mean of the

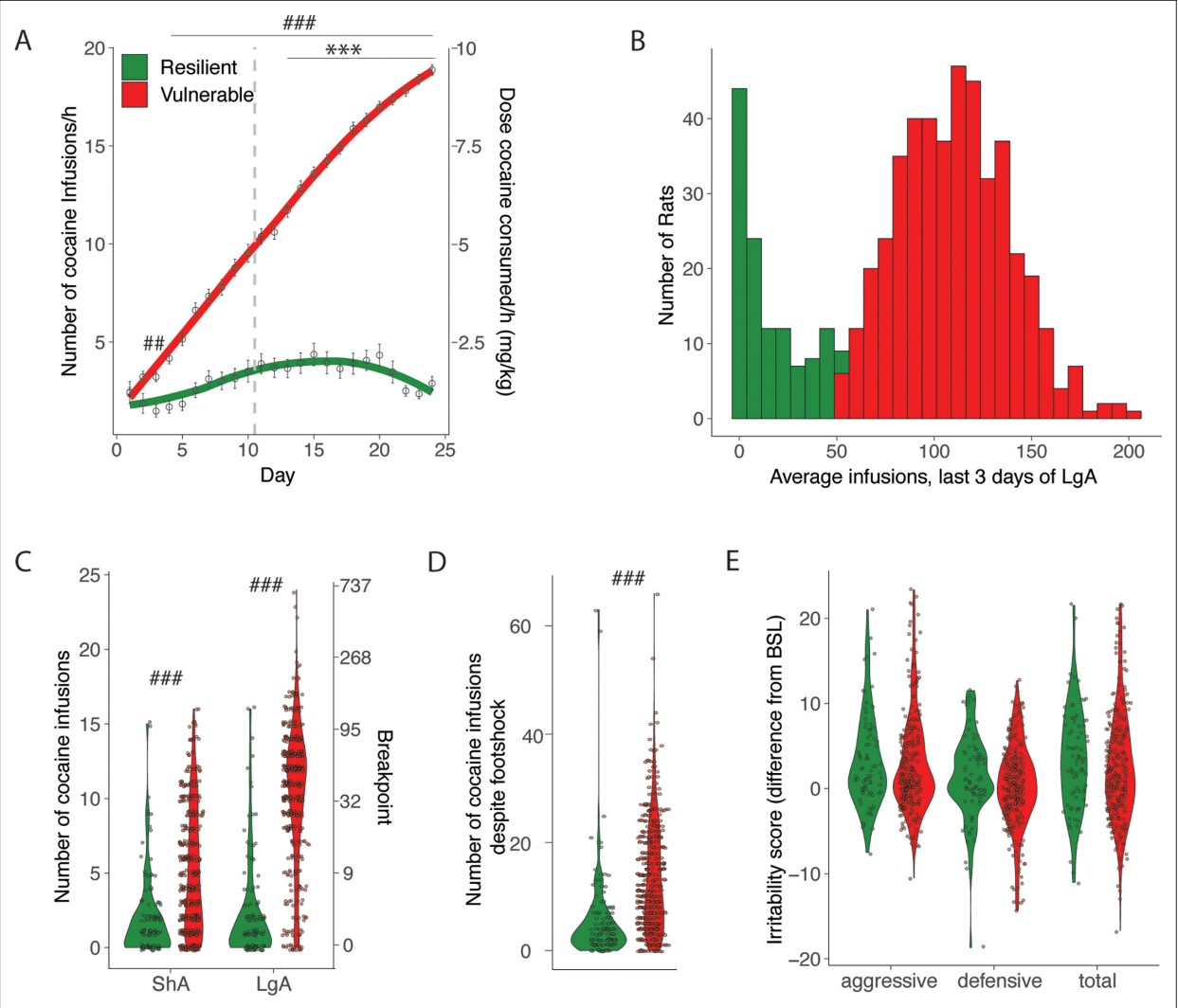

**Figure 3.** Addiction-like behaviors in resilient and vulnerable heterogeneous stock (HS) rats. (**A**) Number of cocaine infusions during short (2 hr, short access: ShA) and long (6 hr, long access: LgA) access of cocaine self-administration in resilient and vulnerable animals (N=122 resilient and 445 vulnerable, ***p<0.001 vs LgA day 1, ###p<0.0001, ##p<0.001 vs resilient. (**B**) Average number of infusions over the last 3 days for the individual animals). (**C**) Number of cocaine infusions under progressive ratio (PR) at the end of ShA and LgA (### p<0.0001 vs resilient). (**D**) Number of infusions despite footshock after LgA (###p<0.0001 vs resilient. (**E**) Irritability scores after LgA (N=79 resilient, 301 vulnerable).

The online version of this article includes the following source data and figure supplement(s) for figure 3:

**Source data 1.** Raw data utilized for generating *Figure 3*.

**Figure supplement 1.** Sex-specific analysis of motivation and compulsivity in vulnerable and resilient rats.

**Figure supplement 1—source data 1.** Raw data utilized for generating *Figure 3—figure supplement 1*.

cohort per sex, and σ is the standard deviation of the cohort per sex. We thus obtained an Escalation Index, Motivation Index, Compulsivity Index, and Irritability Index. *Figure 4A–D* shows the z-scores for escalation, motivation, compulsive-like responding, and irritability-like behavior. A PCA incorporating the z-scores of the 4 behavioral measures showed that almost half of the variability (48%) can be explained by a single PC, which identified vulnerable and resilient rats and aligned with the escalation, motivation, and compulsivity z-scores (eigenvalue >1, explaining 48% of variance and to which all three behaviors contributed in a valuable way, r=0.52–0.60, coefficients 0.6, 0.6, and 0.5, respectively, *Figure 4E*). No difference was identified between the sexes (*Figure 4—figure supplement 1*). The irritability z-score was almost exactly perpendicular to it, indicating that there was little to no correlation with the other addiction-like behavioral measures (r=0.008, coefficient 0.1). For this reason, we

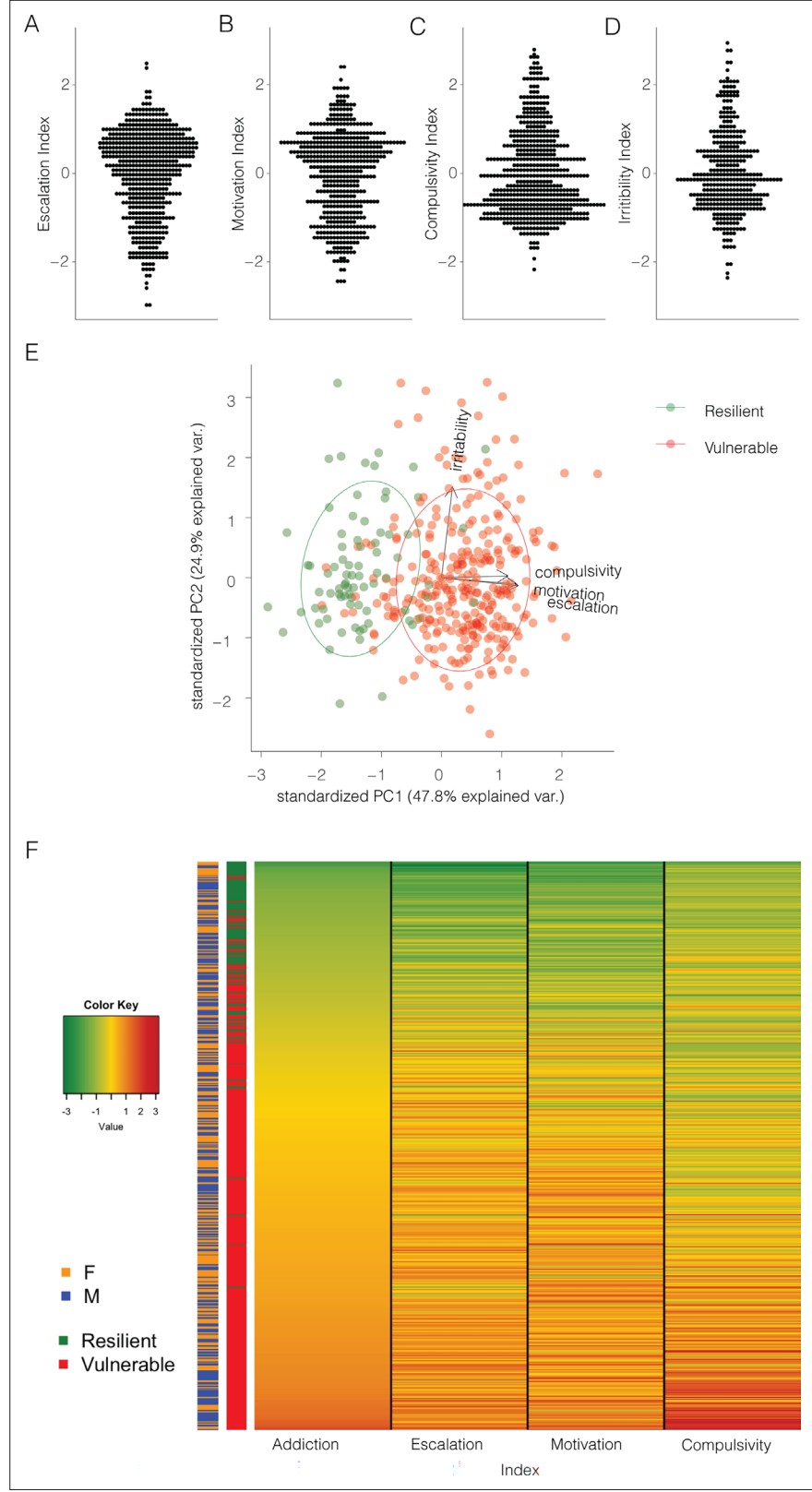

**Figure 4.** Normalizing and combining addiction-like behaviors into an addiction index. (**A**) Z-score for escalation (N=566), (**B**) motivation (N=512), (**C**) compulsivity (N=567), and (**D**) irritability-like behavior (N=380) in the whole population. (**E**) Representation of the individual rats (N=377), resilient (green), or vulnerable (red) along the two first principal components, based on escalation, motivation, compulsivity, and irritability z-scores. (**F**) Representation

*Figure 4 continued on next page*

*Figure 4 continued*

of the addiction index for the individual rats with the constituting individual z-scores and their identification as resilient or vulnerable and male or female (N=511).

The online version of this article includes the following source data and figure supplement(s) for figure 4:

**Source data 1.** Raw data utilized for generating *Figure 4*.

**Figure supplement 1.** Additional principal component analysis (PCA) analysis with the representation of the individual rats (N=377) along the first two principal components.

**Figure supplement 1—source data 1.** Raw data utilized for generating *Figure 4—figure supplement 1*.

calculated an Addiction Index, that provides a comprehensive evaluation of compulsive cocaine use, by averaging the Z-scores of the three dependent variables that explain almost 50% of the variance, escalation, motivation, and compulsive-like behavior, leaving out irritability-like behavior (*Figure 4F*).

## Different degrees of addiction-like behavior

We next used the addiction index to further differentiate between low, mild, moderate, and severe addiction-like behavior in the population, by dividing them into four quartiles. As the indices were derived per sex, quantile normalization resulted in a roughly equal number of males and females in

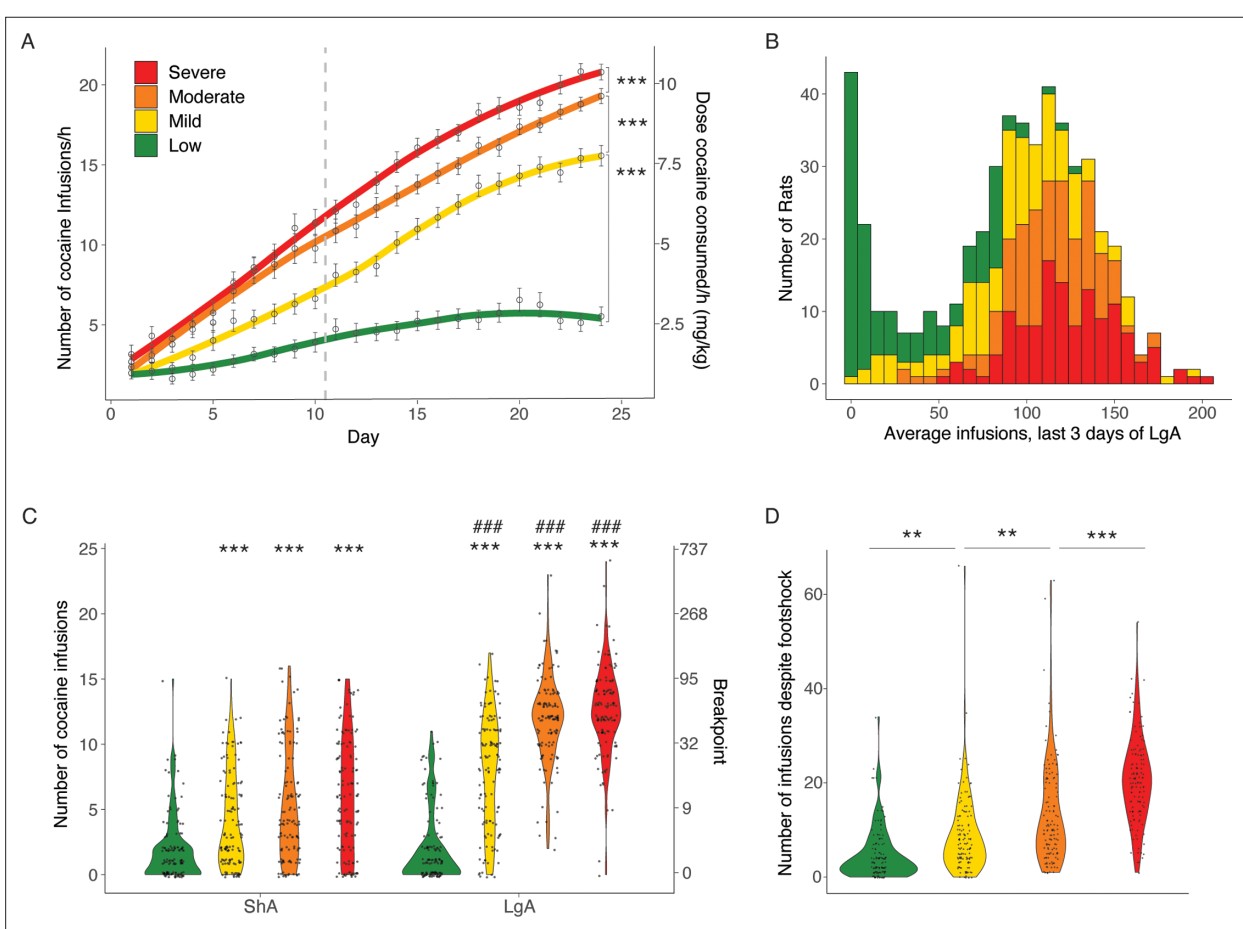

**Figure 5.** Different degrees of vulnerability to cocaine addictive behaviors. (**A**) Cocaine infusions during short (2 hr, short access: ShA) and long (6 hr, long access: LgA) access of cocaine self-administration (N=128 Low + 128 Mild + 127 Moderate + 128 Severe; ***p<0.0001). (**B**) Average number of cocaine infusions over the last 3 days for the individual animals in the low, mild moderate, and severe vulnerable groups. (**C**) Number of cocaine infusions under progressive ratio (PR) test at the end of the ShA and LgA phases (***p<0.0001 vs low, ###p<0.0001 vs ShA) (**D**) Number of infusions despite footshock after LgA for the resilient and vulnerable groups (**p<0.001 and ***p<0.0001).

The online version of this article includes the following source data for figure 5:

**Source data 1.** Raw data utilized for generating *Figure 5*.

each group: 57 females and 71 males in the Low group, 68 females and 60 males in the Mild group, 67 females and 60 males in the Moderate group, and 57 females and 71 males in the Severe group. The intake in the different groups of rats, divided according to their Addiction Index over the last 3 days of LgA is shown in *Figure 5A and B*. Cocaine intake was increased from Low over Mild and Moderate to Severe groups. The two-way ANOVA with a group as between factor and sessions as within factor showed a significant effect of group ($F_{(3, 511)}$=91.36; p<0.0001), sessions ($F_{(23, 511)}$=329.4; p<0.0001) and of the interaction group*sessions ($F_{(69, 11502)}$=18.48; p<0.0001). Pairwise comparisons for the significant main effects of groups, demonstrated that each of the subgroups (divided based on their Addiction Index) obtained more cocaine than their immediate lower subgroup (Severe > Moderate > Mild > Low, p<0.0001; *Figure 5A*). At LgA14 the effect sizes are large between Low and Mild, Moderate and Severe, and between Mild and Severe (Cohen d=1.44 [1.17;1.72], 2.31 [1.99;2.63], 2.50 [2.17;2.83], and 0.81 [0.55;1.07], respectively), medium between Mild and Moderate (Cohen d=0.59 [0.34;0.84]), and small between Moderate and Severe (Cohen d=0.28 [0.03;053]). In the progressive ratio test (*Figure 5C*), the two-way ANOVA effect of group ($F_{(3, 504)}$=176.49; p<0.0001), time ($F_{(1, 504)}$=423.32; p<0.0001) and of the interaction group*time ($F_{(3, 504)}$=52.4; p<0.0001). This was associated with large effect sizes for the Severe, Moderate, and Mild groups (Cohen d=1.65 [1.36;1.93], 1.74 [1.45;2.03], and 0.95 [0.69;1.21], respectively), and a negligible effect size in the Low group Cohen d=0.08 [-0.17;0.32]. Pairwise comparisons showed that the Mild, Moderate, and Severe groups, but not the Low group, showed increased breakpoint after long access, compared to after short access (p<0.0001 after Bonferroni post hoc). Within each time point, the Mild, Moderate, and Severe subgroups showed increased motivation for cocaine to the Low subgroup (p<0.0001 after Bonferroni post hoc, Cohen d=0.71 [0.42;0.96], 0.95 [0.69;1.22], and 1.16 [0.89;1.43] for ShA, or 1.57 [1.29;1.85], 3.20 [2.83;3.58], and 3.36 [2.98;3.74] for LgA, respectively; *Figure 5C*). For the compulsivity test the one-way ANOVA showed a significant main effect ($F_{(3, 507)}$=76.6; p<0.0001) with each of the subgroups showing increased compulsivity compared to their immediate subgroup (p<0.001 after the Bonferroni post hoc test; *Figure 5D*). The associated effect sizes were medium from Low to Mild (Cohen d=0.58 [0.33,0.83]), small from Mild to Moderate (Cohen d=0.45 [0.20,0.70]), and medium from Moderate to Severe (Cohen d=0.72 [0.47,0.98]); with a large effect size between Low and Severe (Cohen d=2.08 [1.77,2.38]). These results demonstrate the variability in the addiction-like behaviors between the individuals, resulting in large effect sizes when comparing the two extreme quartiles Low and Severe (Cohen d~2.5–3.5).

## Discussion

We characterized addiction-like behavior in 556 genetically diverse HS rats by establishing a large behavioral screening aimed at studying different cocaine-related behaviors (escalation, motivation, compulsive-like responding, and irritability-like behavior). A bimodal distribution was observed, with ~20% of rats maintaining low intake levels and ~80% escalating their cocaine intake when given long access. Females exhibited faster acquisition and higher levels of cocaine self-administration compared to males. Individual differences were found, with some rats categorized as resilient and others as vulnerable to addiction-like behaviors. Vulnerable rats showed an increased escalation of cocaine intake, motivation, and compulsive-like responding. Correlational and principal component analysis revealed that a single construct explained 48% of the behavioral variability, separating vulnerable and resilient rats. This construct included the variables associated with escalation of intake, breaking point under a progressive ratio, and continued responding despite adverse consequences under long access, but not irritability-like measures or self-administration measures under short access. Moreover, responding before the shock session strongly predicted continued responding despite adverse consequences during the shock session. The use of an Addiction Index calculated based on the escalation of intake, breaking point under a progressive ratio, and continued responses despite adverse consequences allows us to further identify different levels of severity of addiction-like behaviors from animals exhibiting resilience, low, moderate, and severe addiction-like behaviors.

The high throughput behavioral paradigm used a model of extended access to intravenous self-administration combined with behavioral characterization of compulsivity using progressive ratio responding and responding despite adverse consequences (contingent foot shocks). We chose the model of extended access to cocaine self-administration because it is highly relevant to cocaine use disorder (*Edwards and Koob, 2013*; *George et al., 2014*), and is associated with neuroadaptations

that are also observed in humans with cocaine use disorder (*Adinoff et al., 1990*; *Briand et al., 2008*; *George et al., 2008*; *George et al., 2012*; *Vendruscolo et al., 2012*). After escalation, intake was bimodally distributed. While ~80% of rats (vulnerable) escalated their cocaine intake after being given long access to cocaine,~20% of rats (resilient) maintained a low and stable level of cocaine intake. The low and stable level of intake in resilient rats was not due to lack of catheter patency or failure to self-administer a reward (*Sedighim et al., 2021*).

It is important to note that while our study focused on the differences between resilient and vulnerable rats under long access conditions, the short access period may also be predictive of addiction-like behaviors, particularly in genetically diverse populations. The observed escalation of drug intake during short access in our study is not due to an acquisition issue, as rats start differentiating the active from the inactive levers on the first day of ShA1 (*Figure 1—figure supplement 1B*), with a 3–1 ratio between active/inactive pressing by ShA7. Rather, this early escalation in some individuals could be due to the genetic diversity of the HS rat population, which includes 8 different strains as founder parents. Previous studies have shown profound strain differences in vulnerability to cocaine self-administration and escalation of intake (*Freeman et al., 2009*; *Kosten et al., 2007*; *Perry et al., 2006*; *Picetti et al., 2010*; *Valenza et al., 2016*). Additionally, our use of a 2 hr short access period, compared to the more common 1 hr period, may have allowed for the detection of these individual differences that might otherwise be masked by a ceiling effect in shorter sessions.

The identified resilient group was enriched for males, as over 2/3 of the resilient animals were male. Moreover, we found that females not only showed a low percentage of resilient animals but also showed higher levels of cocaine intake, motivation, and aversion resistance than males in rats with moderate and severe addiction-like behaviors. The increased motivation and compulsive-like responses in females cannot simply be explained by the higher intake among females, as the significant difference remained when we compared only the vulnerable rats in both sexes (see *Figure 3—figure supplement 1*). Previous preclinical studies have reported mixed results on the detection of sex differences in addiction (*Algallal et al., 2020*; *Becker, 2016*; *Roth and Carroll, 2004*; *Swalve et al., 2016*; *Templeton et al., 2023*). In humans, cocaine use disorder is more prevalent in males than in females (*SAMHSA, 2019*), but this might be driven by societal factors. Women may actually be more vulnerable to the reinforcing effects of cocaine and acquire cocaine use disorder faster and at higher levels than males (*Cummings et al., 2011*; *Jackson et al., 2006*). Sex differences were found in HS rats for cocaine cue preference (*King et al., 2021*). The current study was able to replicate the presence of sex differences in cocaine addiction-like behaviors in this large and diverse population with small to moderate effect sizes. The smaller effect size of sex differences, compared to the large effect of drug intake, could explain the contradictory results in the field from underpowered studies. Indeed, a statistical power analysis using every significant sex difference observed in our study shows that studies would require 34–100 animals to find significant sex effects at alpha = 0.05 with 80% power; unfortunately, such sample sizes are rarely used in the field.

To further investigate potential sex differences in the behavioral constructs underlying addiction-like behaviors, we first examined male vs. female differences in the biplot represented in *Figure 4E*, which included the irritability index. This analysis, shown in *Figure 4—figure supplement 1B*, revealed no apparent sex differences in the overall structure of addiction-like behaviors. However, when we performed separate PCAs for males and females (*Figure 4—figure supplement 1C, D*), we found that the relationship between the Compulsivity Index and the other addiction-like behaviors differed between sexes. In males, compulsivity was more positively correlated with irritability, while in females, the relationship was the opposite (*Figure 4—figure supplement 1C, D*). However, when the irritability index was excluded from the analysis, the resulting PCA plots for males and females were almost identical (*Figure 4—figure supplement 1E, F*). These findings suggest that sex differences in the relationship between compulsivity and other addiction-like behaviors may be driven, in part, by differences in irritability. This observation highlights the importance of considering sex-specific patterns when investigating the complex relationships between different aspects of addiction-like behavior.

Withdrawal-induced irritability-like behavior was increased in rats with a history of cocaine self-administration compared to naïve aged-matched rats. This result is in line with previous evidence of a negative emotional state in the long-access model (*Ahmed and Koob, 2005*; *Deroche-Gamonet et al., 2004*; *Mantsch et al., 2004*). However, this increase in irritability-like behavior did not correlate with any self-administration measures and was orthogonal to them in the PCA. Animals taking more

cocaine were not typically the ones that were more irritable, suggesting that irritability may not be a direct measure of withdrawal severity. Similarly, more irritable animals did not necessarily take more cocaine, indicating that irritability may not be a strong driver of escalation in this model, although it may contribute to relapse and craving. The identified increased negative emotional state may thus be independent, capturing a different aspect of addiction, or be due to non-specific effects associated with the protocol, including surgery, chronic catheter implant, and daily testing for approximately 2 months.

Despite the lack of correlation between irritability-like behavior and drug intake in our study, it is important to consider the translational relevance of irritability in the context of substance use disorders. In individuals with a history of substance use disorder, negative affective states, such as irritability, are thought to contribute to continued drug use and relapse through negative reinforcement processes (*Baker et al., 2004*; *Koob and Le Moal, 2008*). Specifically, the desire to alleviate or escape from these unwanted behavioral states may drive individuals to seek and use drugs, thus perpetuating the cycle of addiction (*Baker et al., 2004*; *Solomon and Corbit, 1974*). While our study focused on the relationship between irritability-like behavior and initial drug-seeking behavior, future research should investigate the degree to which irritability acts as a negative reinforcer in the context of drug relapse.

For the purpose of the addiction index, the irritability-like behavior measure was excluded to capture and maximize the diversity associated with PC1's cocaine intake following escalation under progressive ratio and despite aversive footshock. This decision was based on the lack of correlation between irritability and drug intake measures. It is important to note that for the PCA presented in *Figure 4E*, we carefully selected only four key variables: the Z-scores for escalation (average intake at the end of LgA), motivation (intake under progressive ratio), compulsivity (continued responding despite adverse consequences), and irritability. This approach minimized variable redundancy and focused on essential measures of addiction-like behaviors. Furthermore, to ensure the robustness of our findings, we conducted an additional PCA excluding the irritability index, using only the escalation, motivation, and compulsivity indices for the same 377 animals. The results of this analysis (*Figure 4—figure supplement 1A*) corroborated our original findings, with these three variables loading onto a single factor that explained a significant portion of the variance in addiction-like behaviors. However, further studies are needed to differentiate between the possibilities that the increased negative emotional state may be an independent aspect of addiction or due to non-specific effects associated with the protocol, and to investigate the specific role of irritability-like behavior in the context of cocaine use disorder.

Addiction is commonly described as a chronic, recurring condition characterized by excessive drug consumption, compulsive drug seeking, and continued drug use despite negative consequences (*SAMHSA, 2019*). There is a prevailing notion in preclinical addiction research that the only way to identify addiction-like behaviors in individuals is by measuring compulsive drug use and seeking using continued responses despite adverse consequences. Continued drug use or seeking in the face of punishment, such as painful footshocks or aversive tastes, is widely considered the behavioral hallmark of compulsive-like drug-seeking behavior in preclinical models (*Chen et al., 2013*; *Domi et al., 2021*; *Giuliano et al., 2019*; *Li et al., 2021*; *Siciliano et al., 2019*; *Timme et al., 2022*). A key argument in favor of this hypothesis is that several studies have found that drug intake in the absence of an adverse consequence does not predict drug intake in the presence of an adverse consequence, suggesting that they are independent behavioral measures and that only measures of drug taking/ seeking in the context of punishment can measure compulsive-like drug taking/seeking (*Chen et al., 2013*; *Domi et al., 2021*; *Giuliano et al., 2019*; *Li et al., 2021*; *Siciliano et al., 2019*; *Timme et al., 2022*). However, the results from the present report provide strong evidence that in heterogeneous outbred rats, continued responding despite adverse consequences is not an independent measure of compulsive-like responding that is orthogonal to other measures such as escalation of intake and motivation under a progressive ratio schedule of reinforcement. Both correlational and principal component analyses demonstrate that these three behaviors are highly correlated and load onto the same underlying construct.

It is important to consider that the observed correlations between escalation of intake, increased motivation under progressive ratio, and responding despite negative consequences may be influenced by individual differences in baseline response rates. Previous research has shown that animals

with higher baseline rates of responding tend to be less sensitive to punishment and exhibit higher levels of responding under progressive ratio schedules (*Dews, 1955*; *Sanger and Blackman, 1976*). While our findings suggest that these behaviors can be explained by a single psychological construct related to addiction vulnerability, we cannot rule out the possibility that individual differences in baseline response rates may contribute to the observed correlations. Future studies should investigate the relationship between baseline response rates and addiction-like behaviors to further clarify the underlying mechanisms.

A possible explanation as to why previous studies failed to observe this correlation between escalation, motivation, and aversion-resistance is that most of the previous studies used small sample sizes that may not provide sufficient statistical power to observe this relationship between variables. Another explanation is that previous studies often used animal models with limited access to the drug, where animals exhibit low levels of acute intoxication and very few, if any, signs of drug dependence (*George et al., 2022*). However, it is important to note that several behavioral and pharmacological studies have indicated that different measures may capture, at least to some degree, different aspects of addiction-like behavior in alcohol and opioid-dependent rodents (*Aoun et al., 2018*; *Barbier et al., 2015*; *Marchette et al., 2023*). While the present results suggest that escalation of drug intake highly predicts drug response despite adverse consequences in an animal model with long access to cocaine and evidence of drug dependence, further research is needed to determine the extent to which these findings generalize to other drugs of abuse and different stages of the addiction cycle.

It is important to acknowledge that several experimental factors could influence the outcomes of the PCA analysis. These factors include the schedule of reinforcement, the progression of the progressive ratio schedule, the shock intensity, the contingency of the shock, the cocaine unit dose, and the use of multiple punishment sessions (*Belin et al., 2008*; *Deroche-Gamonet et al., 2004*; *Pelloux et al., 2007*). In particular, learning effects may play a role when animals undergo multiple punishment or progressive ratio sessions. An animal's response to punishment or its performance in progressive ratio sessions may change over time as it learns from its previous experiences (*Marchant et al., 2013*; *Vanderschuren et al., 2017*). While the present study utilized a large dataset obtained from a particular experimental design, it is essential to acknowledge that not finding differences in one dataset does not necessarily mean that these differences do not exist. Future studies should investigate the impact of these experimental factors, including learning effects, on the relationship between escalation, motivation, and aversion-resistance to further elucidate the underlying constructs of addiction-like behaviors.

Finally, this report demonstrates the feasibility of characterizing a very large population of heterogeneous rats for addiction-like behavior using an animal model of extended access to intravenous drug self-administration. Such a large-scale behavioral approach is required to unveil differences with smaller effect sizes like sex differences and the genetic basis of addiction-like behavior. The evaluation of different addiction-like behaviors is important, as previous research has suggested that multiple elements of addiction vulnerability may be independently heritable (*Eid et al., 2019*). While our current findings indicate that escalation, motivation, and compulsivity are highly correlated and load onto a single construct in our model, it is possible that distinct genes contribute to different aspects of addiction vulnerability. The high correlation between these behaviors in our study may reflect common underlying genetic influences, but it does not preclude the existence of additional, unique genetic factors that shape specific aspects of addiction-like behavior. Further research is needed to identify the specific genes that contribute to the overall construct of addiction vulnerability, as well as those that may influence distinct behavioral elements. The behavioral characterization of HS rats in this study provides a foundation for future genome-wide association studies (GWAS) aimed at identifying specific alleles and genes that contribute to vulnerability and resilience to cocaine addiction-like behavior (*Chitre et al., 2020*). Moreover, biological samples from animals that are resilient or have mild, moderate, or severe addiction-like behaviors are currently available through the Cocaine Biobank (https://www.drugaddictionresearch.org/) to facilitate collaboration and investigation on the biological mechanisms of addiction-like behaviors (*Carrette et al., 2021*).

In summary, this study examined addiction-like behaviors in a large population of genetically diverse male and female rats. We found that ~80% of the rats were vulnerable and escalated their cocaine intake, while ~20% was resilient and maintained low stable levels. The resilient group had two times more males than females. Females also had higher susceptibility to cocaine self-administration than

males, both qualitatively and quantitively. Correlational and principal component analyses showed that escalation of intake, aversion-resistant responding, and breaking points are highly correlated measures of the same construct in animals with extended access to the drug. This study provides an in-depth characterization of heterogeneous rats with a variety of addiction-like profiles from resilience to mild, moderate, and severe addiction-like behaviors that can be used to investigate the biological mechanisms of addiction.

## Materials and methods

Detailed procedures associated with the experimental timeline in *Figure 1A* can be found in the George lab protocol repository on protocols.io (https://www.protocols.io/workspaces/george-lab).

### Animals

HS rats (Rat Genome Database NMcwiWFsm:HS #13673907, sometimes referred to as N/NIH:HS) were created to encompass genetic diversity by outbreeding eight inbred rat strains (ACI/N, BN/SsN, BUF/N, F344/N, M520/N, MR/N, WKY/N, and WN/N) in 1984 (*Hansen and Spuhler, 1984*; *Solberg Woods and Palmer, 2019*). HS rats were bred at Wake Forest University School of Medicine by Dr. Leah Solberg Woods (n=600). To minimize inbreeding and control genetic drift, each generation of the HS rat colony consists of at least 64 breeder pairs and is maintained using a breeding strategy that minimizes inbreeding, with each breeder pair contributing one male and one female to subsequent generations. Each rat received a chip with an RFID code that was used to track animals throughout the experiment. Rats were shipped at 3–4 weeks of age, kept in quarantine for 2 weeks, and then housed two per cage on a 12 hr/12 hr reversed light/dark cycle in a temperature (20–22°C) and humidity (45–55%) controlled vivarium with ad libitum access to tap water and food pellets (PJ Noyes Company, Lancaster, NH, USA). Animals were tested in 12 cohorts of 46–60 rats per cohort, six cohorts at The Scripps Research Institute, and 6 cohorts at UC San Diego. All procedures were conducted in strict adherence to the National Institutes of Health Guide for the Care and Use of Laboratory Animals and were approved by the Institutional Animal Care and Use Committees of The Scripps Research Institute and UC San Diego (S19016).

### Drugs

Cocaine HCl (National Institute on Drug Abuse, Bethesda, MD) was dissolved in 0.9% sterile saline and administered intravenously at a dose of 0.5 mg/kg/infusion. This dose was selected based on our and other's previous literature demonstrating that this dose is commonly used in rat self-administration studies and is effective in producing addiction-like behaviors (*de Guglielmo et al., 2017*, *Kallupi et al., 2022*, *Kononoff et al., 2018*, *Sedighim et al., 2021*; *Ahmed and Koob, 1998*; *Deroche-Gamonet et al., 2004*; *Belin et al., 2009*). This dose has been shown to maintain stable responding and induce escalation of intake, motivation, and compulsive-like responding in a significant proportion of animals (*Ahmed and Koob, 1998*; *Deroche-Gamonet et al., 2004*; *Belin et al., 2009*). To ensure consistent dosing, animals were weighed weekly to adjust the drug solution concentration, rounded to the nearest ten grams.

### Intravenous catheterization

Rats were anesthetized with vaporized Isoflurane (1–5%). Intravenous catheters were aseptically inserted into the right jugular vein using the procedure described previously (*Kallupi et al., 2020*). Catheters consisted of Micro-Renathane tubing (18 cm, 0.023-inch inner diameter, 0.037-inch outer diameter; Braintree Scientific, Braintree, MA, USA) attached to a 90 degree angle bend guide cannula (Plastics One, Roanoke, VA, USA), embedded in dental acrylic, and anchored with mesh (1 mm thick, 2 cm diameter). Tubing was inserted into the vein following a needle puncture (22 G) and secured with a suture. The guide cannula was punctured through a small incision on the back. The outside part of the cannula was closed off with a plastic seal and metal cover cap, which allowed for sterility and protection of the catheter base. Flunixin (2.5 mg/kg, s.c.) was administered as analgesic, and Cefazolin (330 mg/kg, i.m.) as antibiotic. Rats were allowed three days for recovery prior to any self-administration. They were monitored and flushed daily with heparinized saline (10 U/ml of heparin sodium; American Pharmaceutical Partners, Schaumberg, IL, USA) in 0.9% bacteriostatic sodium

chloride (Hospira, Lake Forest, IL, USA) that contained 52.4 mg/0.2 ml of Cefazolin. Catheter patency was tested throughout and at the end of LgA sessions using a short-acting anesthetic (Brevital), any animal that failed to react to the Brevital infusion was excluded from the study.

## Behavioral testing

### Operant self-administration

Self-administration (SA) was performed in operant conditioning chambers (29 cm × 24 cm×19.5 cm; Med Associates, St. Albans, VT, USA) that were enclosed in lit, sound-attenuating, ventilated environmental cubicles. The front door and back wall of the chambers were constructed of transparent plastic, and the other walls were opaque metal. Each chamber was equipped with two retractable levers that were located on the front panel. Each session was initiated by the extension of two retractable levers into the chamber. Cocaine (0.5 mg/kg per infusion) was delivered through plastic catheter tubing that was connected to an infusion pump. The infusion pump was activated by responses on the right (active) lever that were reinforced on a fixed ratio (FR) 1 schedule, with the delivery of 0.1 mL of the drug per lever press over 6 s followed by a 20 s timeout period that was signaled by the illumination of a cue light above the active lever, during which active lever presses did not result in additional infusions. Responses on the left inactive lever were recorded but had no scheduled consequences. Fluid delivery and behavioral data recording was controlled by a computer with the MED-PC IV software installed. Initially, rats were trained to self-administer cocaine in 10 short access (ShA) sessions (2 hr/day, 5 days/week). At the end of the ShA, phase the animals then underwent 14 extended access (LgA) sessions (6 hr/day, 5 days/week) in order to measure escalation of drug intake.

### Progressive ratio testing

Rats were tested on a PR schedule of reinforcement at the end of each phase (twice after LgA, see *Figure 1* for a detailed timeline), in which the response requirements for receiving a single reinforcement increased according to the following 1, 2, 4, 6, 9, 12, 15, 20, 25, 32, 40, 50, 62, 77, 95, 118, 145, 178, …. The breakpoint was defined as the last ratio attained by the rat prior to a 60 min period during which a ratio was not completed, which ended the experiment.

### Compulsive-like responding using contingent foot shock

A 1 hr footshock punishment testing was conducted between progressive ratio tests following the same parameters as the cocaine self-administration session. Punishment testing followed the same FR1 20 s timeout reinforcement schedule with contingent footshock (0.3 mA, 0.5 s) paired with 30% of the cocaine infusions. This was compared to a 1 hr FR1 20 s timeout reinforcement schedule without footshock the day before, called preshock.

### Irritability-like behavior (bottle brush test)

The bottle brush test consisted of ten 10 s trials with 10 s intertrial intervals in plastic cages (27 cm × 48 cm×20 cm) with clean bedding. The rats were placed in the back of the cage and a bottle brush was rotated rapidly toward the rat's whiskers. Both aggressive responses (smelling, biting, boxing, following, and exploring the bottle brush) and defensive responses (escaping, digging, jumping, climbing, defecation, vocalization, and grooming) were recorded by three trained observers in real-time. Total aggressive and defensive scores were calculated for each animal based on the average score of the observers. Both aggressive and defensive behaviors were summed to calculate the total irritability score. Irritability-like behavior reflects a composite measure of aggressive *vs.* defensive responses (*Kimbrough et al., 2017*). The test was performed after recovery from surgery (baseline) and in withdrawal 18 hr after the last LgA session.

## Analysis

### Calculation of the indices

Z-scores were used to reduce sex and cohort effects and scale the different behavioral paradigm outputs. They were calculated as follows $Z = (x-\mu)/\sigma$, where x is the raw value, $\mu$ is the mean of the cohort per sex, and $\sigma$ is the standard deviation of the cohort per sex. Z-scores were calculated for the following measures: Escalation index is the Z-score of the average intake of an animal on the last 3

days of LgA; Motivation index is the Z-score of the intake during the PR session after LgA; Compulsivity index is the Z-score of the number of infusions during the session with contingent foot shock; Irritability index is the Z-score of the total irritability score with the baseline scores subtracted. The addiction index (AI) was obtained by averaging the relevant behavioral indexes.

## Statistical analyses

Data were analyzed using Prism 9.0 software (GraphPad, San Diego, CA, USA) and R Studio. Self-administration data were analyzed using repeated-measures analysis of variance (ANOVA) followed by Bonferroni post-hoc tests when appropriate. For pairwise comparisons, data were analyzed using the Student's *t*-test. Cohen d effect sizes and 95% confidence intervals were calculated using the R effsize package. Correlations were calculated using Pearson *r* analysis. PCA was performed with prcomp in the stats package, using centering, and scaling. Missing data due to session failure was imputed as the average from the sessions before and after (<2% of data). The data are expressed as mean ± SEM unless otherwise specified. Values of $p < 0.05$ were considered statistically significant.

## Acknowledgements

This work was supported by National Institutes of Health grants U01DA043799 and P50DA037844 from the National Institute on Drug Abuse and funding from the Preclinical Addiction Research Consortium at UCSD. The authors declare no competing financial interests.

## Additional information

### Funding

| Funder | Grant reference number | Author |
|---|---|---|
| National Institute on Drug Abuse | U01DA043799 | Olivier George |
| National Institutes of Health | P50DA037844 | Abraham A Palmer |

The funders had no role in study design, data collection and interpretation, or the decision to submit the work for publication.

### Author contributions

Giordano de Guglielmo, Conceptualization, Data curation, Formal analysis, Investigation, Methodology, Supervision, Writing – original draft, Writing – review and editing; Lieselot Carrette, Data curation, Formal analysis, Investigation, Writing – original draft, Writing – review and editing; Marsida Kallupi, Brent Boomhower, Lisa Maturin, Dana Conlisk, Sharona Sedighim, Lani Tieu, McKenzie J Fannon, Angelica R Martinez, Nathan Velarde, Dyar Othman, Benjamin Sichel, Jarryd Ramborger, Justin Lau, Jenni Kononoff, Adam Kimbrough, Sierra Simpson, Lauren C Smith, Kokila Shankar, Selene Bonnet-Zahedi, Elizabeth A Sneddon, Alicia Avelar, Sonja Lorean Plasil, Joseph Mosquera, Caitlin Crook, Lucas Chun, Ashley Vang, Kristel K Milan, Paul Schweitzer, Supervision; Molly Brennan, Supervision, Methodology; Bonnie Lin, Beverly Peng, Apurva S Chitre, Oksana Polesskaya, Conceptualization; Leah C Solberg Woods, Resources, Methodology; Abraham A Palmer, Resources, Formal analysis, Visualization; Olivier George, Conceptualization, Data curation, Formal analysis, Funding acquisition, Investigation, Project administration, Supervision, Writing – review and editing

### Author ORCIDs

Giordano de Guglielmo ![ORCID] https://orcid.org/0000-0002-4782-7430
Apurva S Chitre ![ORCID] https://orcid.org/0000-0003-1709-9214
Oksana Polesskaya ![ORCID] https://orcid.org/0000-0003-3024-114X
Abraham A Palmer ![ORCID] https://orcid.org/0000-0003-3634-0747
Olivier George ![ORCID] https://orcid.org/0000-0002-3700-5003

## Ethics

All procedures were conducted in strict adherence to the National Institutes of Health Guide for the Care and Use of Laboratory Animals and were approved by the Institutional Animal Care and Use Committees of The Scripps Research Institute and UC San Diego (IACUC protocol S19026). All surgeries were performed under isofluorane anesthesia, and every effort was made to minimize suffering.

Reviewer #1 (Public review): https://doi.org/10.7554/eLife.90422.3.sa1
Reviewer #2 (Public review): https://doi.org/10.7554/eLife.90422.3.sa2
Author response https://doi.org/10.7554/eLife.90422.3.sa3

# Additional files

## Supplementary files
• MDAR checklist

## Data availability

All behavioral data generated and analyzed during this study are included in source data files accompanying this manuscript. The dataset comprises the numerical data used to generate all figures.

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
