## [Editor Report · eLife Assessment]

This manuscript tackles a significant problem in addiction science: how interdependent are measures of "addiction-like" behavioral phenotypes? The manuscript provides **compelling** evidence that, under these experimental conditions, escalation of intake, punishment-resistant responding, and progressive ratio break points reflect a single underlying construct rather than reflect distinct unrelated measures. The exceptionally large sample size and incorporation of multiple behavioral endpoints add strength to this paper, and make it an **important** resource for the field.

---

## [Referee Report · Reviewer #1 (Public review)]

Summary:

Guglielmo et al. characterized addiction-like behaviors in more than 500 outbred heterogeneous stock (HS) rats using extended access to cocaine self-administration (6 h/daily) and analyzed individual differences in escalation of intake, progressive-ratio (PR) responding, continued use despite adverse consequence (contingent foot shocks), and irritability-like behavior during withdrawal. By principal component analysis, they found that escalation of intake, progressive ratio responding, and continued use despite adverse consequences loaded onto the same factor, whereas irritability-like behaviors loaded onto a separate factor. Characterization of rats in four categories of resilient, mild, moderate, and severe addiction-like phenotypes showed that females had higher addiction-like behaviors, particularly due to a lower number of resilient individuals, than males. The authors suggest that escalation of intake, continued use despite adverse consequences, and progressive ratio responding are highly correlated measures of the same psychological construct and that a significant proportion of males, but not females may be resilient to addiction-like behaviors. The amount of work in this study is impressive, and the results are interesting.

Strengths: Large dataset. Males and females included.

---

## [Referee Report · Reviewer #2 (Public review)]

Summary:

In this paper by de Guglielmo and colleagues, the authors were interested in analyzing addiction-like behaviors using a very large number of heterogeneous outbred rats in order to determine the relationships among these behaviors. The paper used both males and females on the order of hundreds of rats, allowing for detailed and complex statistical analyses of the behaviors. The rats underwent cocaine self-administration, first via 2-hour access and then via 6-hour access. The rats also underwent a test of punishment resistance in which footshocks were administered a portion of the times a lever was pressed. The authors also conducted a progressive ratio test to determine the break point for "giving up" pressing the lever and a bottle-brush test to determine the rats "irritability". Ultimately, principal component analysis revealed that escalation of intake during 6-hour access, punishment resistance, and breakpoint all loaded onto the same principal component. Moreover, the authors also identified a subgroup of "resilient" rats that qualitatively differed from the "vulnerable" rats and also identified sex differences in their work.

Strengths:

The use of heterogeneous rats and the use of so many rats are major strengths for this paper. Moreover, the statistical analyses are particular strengths as they enabled the identification of the three measures as likely reflecting a single underlying construct. The behavioral methods themselves are also strong, as the authors used behavioral measures commonly used in the field that will enable comparison with the field at large. In general, the results support the conclusions and provide a wealth of data to the field. The addition of effect sizes is also a strength, as this provides critical information to other researchers.

Additionally, the changes made to the manuscript are another strength, as the authors clearly took the reviewers' points seriously and made strong efforts incorporate the reviewers' ideas.

The manuscript also uses both males and females and provides a good analysis of how findings differed by sex as well as how large the effect sizes were for those differences.

---

## [Author Response]

The following is the authors’ response to the original reviews.

**Reviewer #1:**
(1) Adding page numbers would have helped the reviewers.

We apologize for the oversight and have added page numbers for the revision.

(2) Page 2, second paragraph: please do not generalize. Also, this sentence is confusing: "the addiction neuroscience field has moved from recognizing that "compulsive drug seeking/use" and "continued seeking/use despite negative consequences" are two distinct aspects of addiction to defining the former nearly exclusively by the latter in animal models."

We acknowledge that the sentence in question may have been unclear. We have revised the introduction to avoid generalizations and improve clarity to read:

“Recently, the preclinical addiction field has moved from recognizing compulsive drug seeking/use and continued seeking/use despite negative consequences as two distinct aspects of addiction, to examining compulsive-like behavior nearly exclusively by models of continued seeking/use despite negative consequences.”

In the revised introduction, we have focused on the specific aims and findings of our study, emphasizing the use of a large, genetically diverse sample and an extended drug access paradigm to better model addiction-like behaviors. We have also clarified the relationship between the different measures of addiction-like behavior and the potential role of sex differences in resilience to these behaviors.

(3) Again here, please do not generalize: "While these three behaviors capture different aspects of addictionlike behaviors, a pervasive view in the field is that the only way to identify an individual with an addiction phenotype is to measure continued drug use despite adverse consequences." This is not unanimous in the addiction field. Same on page 21.

We have revised the sentence to avoid generalizing and to acknowledge that this perspective is held by some researchers, rather than presenting it as a pervasive view. We have also included relevant citations to support this point. This sentence now reads:

“These measures are thought to capture different aspects of addiction-like behaviors. Some researchers argue that continued drug use despite adverse consequences is the most critical measure for identifying an addiction phenotype, as it reflects the compulsive nature of drug use (Deroche-Gamonet et al., 2004; Vanderschuren and Everitt, 2004)”

(4) This sentence needs citations: "A key argument in favor of this hypothesis is that responding despite adverse consequences is sometimes uncorrelated to drug taking/seeking."

We have added references (Chen et al., 2013; Domi et al., 2021; Giuliano et al., 2019; Li et al., 2021; Siciliano et al., 2019; Timme et al., 2022; Belin et al., 2008; Pelloux et al., 2007) that provide evidence for this assertion. These studies demonstrate that individual differences in responding despite adverse consequences can be dissociated from drug intake and seeking behaviors, suggesting that they may measure distinct aspects of addiction-like behaviors.

(5) Page 4: what is "an advanced model?" (also on page 22). Change "characterization" to "characterized." Delete "as much as possible" in "as much genetic diversity as possible."

These have been addressed

(6) Page 7, statistical analysis: PCA needs to be explained further. Was the PCA varimax rotated, normalized, eigenvalues, etc. Was this used to find "latent variables?" (PCA versus factor analysis)

It was a principal component analysis (PCA), deriving components that are a linear combination of the original variables, with the following coefficients for the first two components, which were added in the results for PC1:

**Author response table 1. sa3table1:** 

	PC1	PC2
Escalation Index	0.605	-0.089
Motivation Index	0.594	-0.071
Compulsivity Index	0.524	0.020
Irritability Index	0.086	0.993

The PCA was performed in R with prcomp in the stats package, using centering and scaling, which was added in the methods section. No orthogonal loadings rotation (varimax) was used. The eigenvalues of the PCs are 1.9, 1.0, 0.7, 0.4 and explain variance as shown in the scree plot:

(7) Page 9: correct "an indexes."

This was corrected as ‘indexes’

(8) Figure 1 legend: correct "test at the."

Corrected to ‘tested’

(9) Page 17: rewrite "except for the low addicted one."

Done

(10) Page 19: delete "state-of-the-art." Intravenous self-administration is not new.

Done

(11) Page 20: replace "abuse" with cocaine use disorder.

Done

(12) Page 20: The distinction of qualitative and quantitative differences between males and females is inaccurate given that resilient and vulnerable groups were arbitrarily defined by quantitative differences.

This distinction between quantitative and qualitative was removed.

(13) The discussion about DSM-V criteria is "over the top" and unnecessary. One cannot determine whether rodents took more drugs than intended, made efforts to quit, etc.

This discussion was toned down and shortened, as this is not the focus of the manuscript.

(14) Page 21: The discussion about small n and the test of nondependent rats should also be toned down and it is incomplete. There are several behavioral and pharmacological studies that indicate that different measures may capture, at least to some degree, different aspects of behavior in alcohol and opioid-dependent rodents (e.g., PMID: 28461696; PMID: 25878287; PMID: 36683829).

the discussion has been toned down and expanded as suggested by the reviewer.

Now it reads:

“A possible explanation as to why previous studies failed to observe this correlation between escalation, motivation, and aversion-resistance is that most of the previous studies used small sample sizes that may not provide sufficient statistical power to observe this relationship between variables. Another explanation is that previous studies often used animal models with limited access to the drug, where animals exhibit low levels of acute intoxication and very little, if any, signs of drug dependence (George et al., 2022). However, it is important to note that several behavioral and pharmacological studies have indicated that different measures may capture, at least to some degree, different aspects of addiction-like behavior in alcohol and opioid-dependent rodents (Aoun et al., 2018; Barbier et al., 2015; Marchette et al., 2023). While the present results suggest that escalation of drug intake highly predicts drug responding despite adverse consequences in an animal model with long access to cocaine and evidence of drug dependence, further research is needed to determine the extent to which these findings generalize to other drugs of abuse and different stages of the addiction cycle”.

(15) Several factors should be considered for explaining their PCA findings. The progression of the progressive ratio (too steep, not steep enough), the shock intensity (too low, too high), the contingency of the shock (high or not high enough), the cocaine unit dose, the use of multiple punishment sessions (learning; the first session is likely to reflect the previous session, same for PR) etc, all could affect the outcomes. Not finding differences in one dataset (even large ones) obtained from a particular experimental design does not necessarily mean that these differences do not exist.

Thank you for raising this important point about the potential impact of experimental factors on our PCA findings. We now acknowledge in the discussion (pages 22-23) that several factors, such as the progression of the progressive ratio schedule, shock intensity, contingency of the shock, cocaine unit dose, and the use of multiple punishment sessions, could influence the outcomes of our analysis. Now it reads:

“It is important to acknowledge that several experimental factors could influence the outcomes of the PCA analysis. These factors include the schedule of reinforcement, the progression of the progressive ratio schedule, the shock intensity, the contingency of the shock, the cocaine unit dose, and the use of multiple punishment sessions (Belin et al., 2008; Deroche-Gamonet et al., 2004; Pelloux et al., 2007). In particular, learning effects may play a role when animals undergo multiple punishment or progressive ratio sessions. An animal's response to punishment or its performance in progressive ratio sessions may change over time as it learns from its previous experiences (Marchant et al., 2013; Vanderschuren et al., 2017). While the present study utilized a large dataset obtained from a particular experimental design, it is essential to acknowledge that not finding differences in one dataset does not necessarily mean that these differences do not exist. Future studies should investigate the impact of these experimental factors, including learning effects, on the relationship between escalation, motivation, and aversion-resistance to further elucidate the underlying constructs of addiction-like behaviors.”

(16) Related to the above, another reason for all "consummatory variables" to load onto the same factor can be due to the selection of the variables. For example, the inclusion of all ShA and LgA access sessions makes the PCA much less powerful. In fact, these many similar variables would make the PCA less powerful in a large dataset than a much smaller dataset that includes fewer variables in the PCA. The authors should attempt to avoid redundant variables in the PCA (all ShA and all LgA sessions). Perhaps use the average of the last three sessions of each ShA and LgA (or the slope of the escalation curve for LgA), or not even include ShA. They should also attempt PCAs without the irritability test. It is very common to find clusters of variables pertaining to the same tests i.e., all consummatory variables clustered together, and all irritability measures clustered together in an independent factor.

For the PCA in figure 4E, only 4 variables were included: the Z-scores for (A) escalation (calculated as the average intake of the last three long-access sessions, similar to the average or slope of the escalation curve as suggested by the reviewer), (B) motivation (intake under progressive ratio), (C) compulsivity (continued responding despite adverse consequences), and (D) irritability. This approach aimed to minimize redundancy in the variables and focus on key measures of addiction-like behaviors.

To further address the reviewer's concern, we performed an additional PCA on the same 377 animals, excluding the irritability index. This PCA included only the escalation, motivation, and compulsivity indices. The results of this analysis (Figure S3A) were consistent with our original findings, with the three variables loading similarly (>+1 standard deviation) onto factor 1 explaining 63.5% of the variance in addiction-like behaviors." This analysis was added as supplementary figure S3A.

(17) Also related to the above, males and females may behave differently, sometimes in opposite directions, thus "cancel each other out." The authors should take advantage of their huge sample size and do PCAs separately for males and females to learn more about potential sex differences in behavioral constructs.

First, we looked at male vs. female differences in the biplot represented in Fig. 4E, which included the irritability index. This analysis showed no sex differences and was added as supplemental figure S3B.

Next, we ran the PCA analysis on males (left panel) and females (right panel) separately, which revealed a difference in the relationship between the Compulsivity Index and the other variables. In males, the Compulsivity Index separated from the escalation and motivation indices in the opposite direction relative to PC2 compared to females. Additionally, in males, compulsivity became more positively correlated with irritability, while in females, the relationship was opposite. These interpretations were added to the discussion page 21 and the results were included in the Supplemental Figure S3 C-D. The discussion was updated accordingly.

(18) Figure 3 legend. There is no correlation in the figure.

This was intended to summarize that vulnerable animals, as defined with a high intake in the last 3 LgA sessions are also more vulnerable in the other measures, but was removed to avoid further confusion.

(19) Page 22: the authors contradict themselves: "The evaluation of different addiction-like behaviors is important as multiple elements of addiction vulnerability were found to be independently heritable (Eid et al., 2019), and likely controlled by distinct genes that remain to be identified."

we agree with the reviewer, and we edited the discussion to clarify the relationship between the current findings and the potential for distinct genetic influences on different aspects of addiction vulnerability. The text now reads:

“The evaluation of different addiction-like behaviors is important, as previous research has suggested that multiple elements of addiction vulnerability may be independently heritable (Eid et al., 2019). While our current findings indicate that escalation, motivation, and compulsivity are highly correlated and load onto a single construct in our model, it is possible that distinct genes contribute to different aspects of addiction vulnerability. The high correlation between these behaviors in our study may reflect common underlying genetic influences, but it does not preclude the existence of additional, unique genetic factors that shape specific aspects of addiction-like behavior. Further research is needed to identify the specific genes that contribute to the overall construct of addiction vulnerability, as well as those that may influence distinct behavioral elements. The behavioral characterization of HS rats in this study provides a foundation for future genome-wide association studies (GWAS) aimed at identifying specific alleles and genes that contribute to vulnerability and resilience to cocaine addiction-like behavior (Chitre et al., 2020).”

**Reviewer #2**:(1) I strongly suggest the authors include effect sizes. They are likely correct that many studies using rats during self-administration are underpowered, but because it is unlikely that most studies will use over 500 rats, the effect size information would be beneficial for future researchers. That is, if an effect requires 100 rats per group, this would be critical to know.

Standardized effect sizes (Cohen d and 95% confidence intervals) were included for the sex differences, intake group differences, and addiction groups. Moreover, a statement about the required amount of animals needed to detect significant effects was added in the discussion.

(2) I suggest that the authors tone down the portions of the Discussion that appear to be defenses of the extended access model. The data in this paper do not address short vs. long-access in a way that supports that. Moreover, they should acknowledge some of the ways that I noted above in which the short access period seems to be just as predictive as the long-access. It raises the question of whether keeping another group of rats on short access through all 25 days would have led to some of the same outcomes that were observed.

This discussion was toned down and shortened, as this is also not the focus of the manuscript (see also response to reviewer 1’s 14th comment).

We appreciate the reviewer's comment on the potential predictive value of the short access period for addiction-like behaviors. We agree that maintaining a group of rats on short access throughout the 25 days could have provided valuable insights into the development of these behaviors, particularly in light of the individual differences observed in our genetically diverse HS rat population. As we mention also in our response to Reviewer 3 (comment 5), the observed escalation of drug intake during the short access condition in our study may be attributed to the genetic diversity of the HS rat population. To address this important point, we have added a new paragraph in the Discussion section that elaborates on this observation:

"It is important to note that while our study focused on the differences between resilient and vulnerable rats under long access conditions, the short access period may also be predictive of addiction-like behaviors, particularly in genetically diverse populations. The observed escalation of drug intake during short access in our study is not due to an acquisition issue, as rats start differentiating the active from the inactive levers on the first day of ShA1 (Fig. S1B), with a 3 to 1 ratio between active/inactive pressing by ShA7. Rather, this early escalation may be attributed to the individual differences in drug-taking behavior among the HS rats, highlighting the importance of using genetically diverse animals to capture the full spectrum of individual differences in addiction-like behaviors."

(3) I suggest the authors explain how the dosing was maintained across the self-administration period. I also suggest that the authors provide figures that show mg/kg of cocaine consumed for each day, rather than just infusions per day. This would be especially helpful for the sex difference claims.

To ensure consistent dosing, animals were weighed weekly to adjust the drug solution concentration, rounded to the nearest ten grams. This sentence was added to the methods section. Each infusion is 0.5 mg/kg, so the amount the animals consumed = number of infusions x 0.5 mg/kg. Moreover, a second axis with the dose in mg/kg of cocaine consumed was added to the escalation curves in figures 1B, 2A, 3A, 5A, and S1A.

(4) Throughout the paper, and especially the 2nd paragraph of the Introduction, the authors make a number of assertions for which they should provide references.

We have carefully reviewed the manuscript and have now included relevant references to ensure that all statements are properly supported by the existing literature.

(5) Likewise, with the Discussion about sex and gender differences, I suggest a more nuanced and better-cited discussion. Many rodent studies with self-administration have not identified sex differences, though this often gets under-noticed as the titles and abstracts do not mention the lack of effects. The support for gender differences in humans in terms of vulnerability to cocaine use disorder, beyond that men have higher rates, is thin and this section should be modified.

The section was modified with additional references and linked to the newly introduced effect sizes for sex differences.

(6) I also suggest the authors change some of the language such as referring to their behavioral measures as "state of the art". Extended access has been around for over two decades.

This has been adjusted, also see response to reviewer 1’s comments on page 4, 19, and 22.

**Reviewer #3:**
Strengths:(1) The number of animals run through this study is particularly impressive and allows for analyses that cannot be done with smaller cohorts.(2) The inclusion of males and females in a study of this size allows for a better understanding of potential sex differences across a range of behavioral domains.(3) Relating these measures to each other is incredibly important. If they are all measuring the same thing this would have important implications for the field.Weaknesses:(1) The authors claim that escalation of intake, increased motivation under progressive ratio, and responding despite negative consequences can all be explained by the same psychological construct, which they conclude is predictive of an addiction-like phenotype. However, previous research has demonstrated that the aforementioned behavioral measures highly correlate with the rate at which animals lever press to receive a reinforcer. For example, animals that have higher baseline rates of behavior will also be less sensitive to punishment and will press more on a PR. In fact, early behavioral pharmacology work from Peter Dews showed that the same is true for drug effects on behavior, where the same drug has less of a behavioral effect with behavior was maintained on a schedule that resulted in higher response rates. This is not ruled out and actually could explain the results in a parsimonious way. This is not highlighted or mentioned in the manuscript.

Thank you for raising this important point about the potential influence of baseline response rates on the observed correlations between addiction-like behaviors. We agree that individual differences in baseline response rates may contribute to the relationships we observed, and we have added a paragraph to the discussion acknowledging this possibility (see page 22). We now discuss how previous research has shown that animals with higher baseline rates of responding tend to be less sensitive to punishment and exhibit higher levels of responding under progressive ratio schedules, as demonstrated in early behavioral pharmacology work by Dews and others (Dews, 1955; Sanger and Blackman, 1976). While our findings suggest that escalation of intake, motivation, and responding despite negative consequences can be explained by a single psychological construct related to addiction vulnerability, we cannot rule out the influence of baseline response rates. We have highlighted the need for future studies to investigate the relationship between baseline response rates and addiction-like behaviors to further clarify the underlying mechanisms

(2) The authors draw major conclusions from data collected using only one dose of cocaine. Can the authors comment on how the dose of cocaine was selected? Although the majority of the animals maintained responding to the drug, one finding of the manuscript claims that roughly 20% of animals were resilient to developing an addiction-like phenotype. The differences observed could simply be a result of selecting too high or too low of a dose per infusion.

We selected a dose of 0.5 mg/kg/infusion of cocaine for our study based on our and others previous literature demonstrating that this dose is commonly used in rat self-administration studies and is effective in producing addiction-like behaviors (de Guglielmo et al. 2017, Kallupi et al. 2022, Kononoff et al. 2018, Sedighim et al. 2021, Ahmed and Koob, 1998; Deroche-Gamonet et al., 2004; Belin et al., 2009). This dose has been shown to maintain stable responding and induce escalation of intake, motivation, and compulsive-like responding in a significant proportion of animals (Ahmed and Koob, 1998; DerocheGamonet et al., 2004; Belin et al., 2009).

(3) In line with the previous comment, rats self-administered cocaine under one schedule of reinforcement and were exposed to only one, mild, foot shock intensity. Although a large number of animals were used, it is difficult to translate these results to understand patterns of drug intake in humans.

We appreciate the reviewer's comment on the limitations of using a single schedule of reinforcement and a single foot shock intensity in our study. We acknowledge that these factors may limit the direct translatability of our findings to patterns of drug intake in humans. As mentioned in our response to Reviewer 1 (comment 15), we have now added a paragraph to the discussion (pages 22-23) addressing the potential impact of various experimental factors on our PCA findings. These factors include the schedule of reinforcement, the progression of the progressive ratio schedule, shock intensity, contingency of the shock, cocaine unit dose, and the use of multiple punishment sessions. We acknowledge that the specific parameters used in our study may have influenced the observed individual differences in addiction-like behaviors and that different results might be obtained under different experimental conditions. To further address the current reviewer's concern, we would like to emphasize that our study aimed to investigate individual differences in addiction-like behaviors within a specific experimental context, rather than directly modeling the complex patterns of drug intake in humans. While our findings provide valuable insights into the relationship between different addiction-like behaviors in rats, we agree that additional studies using a range of experimental conditions are needed to fully understand the extent to which these findings translate to human drug use patterns. Future studies could investigate the impact of different schedules of reinforcement, shock intensities, and other experimental parameters on the development and expression of addiction-like behaviors in the HS rat population. Such studies would help to determine the generalizability of our findings and provide a more comprehensive understanding of the factors influencing individual differences in addiction vulnerability.

(4) It is unclear how a principal component analysis, which includes irritability-like behavior, was conducted when the total number of animals used for behavior is nearly half the number of animals used for drugintake behaviors. The authors should expand on the PCA methodology and explain how that is not a problem for the PCA method that is used.

The PCA (Figure 4E) can only be performed using animals that had the data for all measures, including irritability. Since not all animals were tested for irritability-like behaviors the PCA was performed on those 377 animals who had behavioral measures for all variables. Once irritability was excluded as a measure, the larger animal set could be used (including the animals missing irritability measurements). This was clarified in the text and figure legend, where animal numbers were added.

(5) It is surprising that the authors observed an escalation of drug intake during the short access condition (Fig. 1B, 2A, 3A, 5A). Previous literature has demonstrated that animals with short access to cocaine maintain stable and low intake, even when tested daily for weeks. Can the authors comment on this discrepancy? Are these animals still acquiring the task during this period?

We were indeed surprised by the fact that some individuals started escalating their intake early on during short access, as most of the literature shows that short access leads to stable intake. However, we have some hypotheses that may explain this phenomenon. It is unlikely that this early escalation is due to an acquisition issue as rats start differentiating the active from the inactive levers on the first day of ShA1 (new data included as Fig. S1B) and that there is a 3 to 1 ratio between active/inactive pressing by ShA7. Three factors are more likely to play a key role in this early escalation. First, it is likely that the early escalation observed in some animals is due to the genetic diversity of the HS rat population used in our study. Indeed, most of the literature used Wistar, Sprague Dawley, and Long Evans rats, while the HS rats includes 8 different strains as founder parents. Indeed, profound strain differences have been observed in the vulnerability to self-administer cocaine, the maintenance of cocaine self-administration during short access, and the level of escalation of intake (Freeman et al., 2009; Kosten et al., 2007; Perry et al., 2006; Picetti et al., 2010; Valenza et al., 2016). Second, we used a 2 h short access while most studies used 1 h of short access. The level of escalation is proportional to the duration of access, and it is likely that a 1 h access period leads to a ceiling effect preventing detection of individual differences in early escalation. Third, it is likely that reporting and publication bias played a significant role in the lack of reporting of such a phenomenon. When using a low sample size, many laboratories remove outliers during short access to ensure a homogeneous population before being given long access or moving on to a specific experimental condition. The combination of using a limited number of strains with limited genetic diversity, a 1h short access, and reporting bias is likely to have led to the conclusion that escalation of cocaine intake does not occur during short access. The current report using a rat stock with high genetic diversity, a 2 h short access, and no reporting bias conclusively demonstrates that escalation of cocaine intake occurs in some individuals. The discussion has been updated to reflect these points on page 20.

(6) Although the authors provide PR and foot shock data separated by sex in Supplemental Figure 2, the manuscript would benefit from denoting the number of males and females in each data set shown in Figures 3 and 5. Is there a difference in the proportion of males or females that display a vulnerable phenotype? Given that the authors are interested in investigating sex differences, it would greatly improve the manuscript to disaggregate the resilient/vulnerable data (Figure 3) and degree of vulnerability data (Figure 5) by sex.

We have now added the proportion of males and females in each of these subgroups and discussed these results.

- For figure 3: when categorizing on intake, there is a greater number of males in the Resilient population than females, as a logical conclusion from the findings in figure 2. The following was added: “From the analysis of sex differences above, we could expect the Resilient group to contain more males. Amongst the resilient animals, there were twice as many males compared to females (N = 122 total with 82 males and 40 females). The amounts in the vulnerable group were almost equal (N = 445 total with 210 males and 235 females).

- For figure 5: as the z-scoring of the behavioral measures is performed per sex, these differences are normalized, and all groups contain equal amounts of males and females. The following was added: “As the indices were derived per sex, quantile normalization results in roughly equal number of males and females in each group: 57 females and 71 males in the Low group, 68 females and 60 males in the Mild group, 67 females and 60 males in the Moderate group, and 57 females and 71 males in the Severe group.” To make this clearer, we also elaborated on the calculation of the indices in the methods and results sections.

(7) Consistent with previous reports, the authors demonstrate an increase in irritability-like behavior during withdrawal after cocaine self-administration; however, they make the claim that this variable was orthogonal to drug intake behavior. The discussion claims that the increase in irritability-like behavior was likely due to factors independent of drug intake, such as undergoing surgery, catheter implants, or being tested daily for two months. Individuals with a history of substance use disorder are thought to continue use as a consequence of negative reinforcement. Unwanted behavioral states, such as irritability, can be a driving factor in relapse; therefore, it would perhaps be more translationally relevant to understand the degree to which irritability-like behavior acts as a negative reinforcer rather than correlating this behavior with initial drug-seeking behavior. While this is outside of the scope of the current manuscript, perhaps this is worth noting in the discussion.

the reviewer raises a good point and we added a paragraph to the discussion acknowledging the translational relevance of understanding the relationship between irritability and drug-seeking behavior in the context of negative reinforcement and relapse. Now it reads:

“Despite the lack of correlation between irritability-like behavior and drug intake in our study, it is important to consider the translational relevance of irritability in the context of substance use disorders. In individuals with a history of substance use disorder, negative affective states, such as irritability, are thought to contribute to continued drug use and relapse through negative reinforcement processes (Baker et al., 2004; Koob and Le Moal, 2008). Specifically, the desire to alleviate or escape from these unwanted behavioral states may drive individuals to seek and use drugs, thus perpetuating the cycle of addiction (Baker et al., 2004; Solomon and Corbit, 1974). While our study focused on the relationship between irritability-like behavior and initial drug-seeking behavior, future research should investigate the degree to which irritability acts as a negative reinforcer in the context of drug relapse”.